# Syndromic surveillance: A key component of population health monitoring during the first wave of the COVID-19 outbreak in France, February-June 2020

Marie-Michèle Thiam[1]*, Isabelle Pontais[1], Cécile Forgeot[1], Gaëlle Pedrono[1], SurSaUD® Regional Focal Point[2¶], SOS Médecins[3¶], Group of Emergency Medicine Doctors[4¶], Louis-Marie Paget[5], Anne Fouillet[1], Céline Caserio-Schönemann[1]

1 Data Science Division, Santé Publique France, Saint-Maurice, France, 2 Regional Division, Santé Publique France, Saint-Maurice, France, 3 SOS Médecins France National Board, Paris, France, 4 OSCOUR® Network, Marseille, France, 5 Non-Communicable and Traumatic Diseases Division, Santé Publique France, Saint-Maurice, France

¶ Membership of SurSaUD® Regional Focal Point, SOS Médecins, and Group of Emergency Medicine Doctors is provided in the Acknowledgments.
* marie-michele.thiam@santepubliquefrance.fr

**Data Availability Statement:** OSCOUR data are available through this URL:

## Abstract

### Background

The French syndromic surveillance (SyS) system, SurSaUD®, was one of the systems used to monitor the COVID-19 outbreak.

### Aim

This study described the epidemiological characteristics of COVID-19-related visits to both emergency departments (EDs) and the network of emergency general practitioners known as SOS Médecins (SOSMed) in France from 17 February to 28 June 2020.

### Methods

Data on all visits to 634 EDs and 60 SOSMed associations were collected daily. COVID-19-related visits were identified using ICD-10 codes after coding recommendations were sent to all ED and SOSMed doctors. The time course of COVID-19-related visits was described by age group and region. During the lockdown period, the characteristics of ED and SOSMed visits and hospitalisations after visits were described by age group and gender. The most frequent diagnoses associated with COVID-19-related visits were analysed.

### Results

COVID-19 SyS was implemented on 29 February and 4 March for EDs and SOSMed, respectively. A total of 170,113 ED and 59,087 SOSMed visits relating to COVID-19 were recorded, representing 4.0% and 5.6% of the overall coded activity with a peak in late March representing 22.5% and 25% of all ED and SOSMed visits, respectively. COVID-19-related

https://geodes.santepubliquefrance.fr/#bbox=-1343527,6775601,3151466,1847697&c=indicator&view=map2 SOS Médecins data are available through this URL: https://geodes.santepubliquefrance.fr/#bbox=-1343527,6775601,3151466,1847697&c=indicator&view=map2.

**Funding:** The author(s) received no specific funding for this work.

**Competing interests:** The authors have declared that no competing interests exist.

visits were most frequently reported for women and those aged 15–64 years, although patients who were subsequently hospitalised were more often men and persons aged 65 years and older.

## Conclusion

SyS allowed for population health monitoring of the COVID-19 epidemic in France. As SyS has more than 15 years of historical data with high quality and reliability, it was considered sufficiently robust to contribute to defining the post-lockdown strategy.

## Introduction

The first cases of the novel coronavirus disease 2019 (COVID-19) were reported in Wuhan, China, in December 2019 [1]. On 30 January 2020, the World Health Organisation (WHO) declared COVID-19 as a public health emergency of international concern [2]. In France, the first cases were confirmed on 24 January 2020. A national surveillance strategy was gradually implemented by Santé publique France (SpFrance), the French Public Health Agency, from 13 January 2020 [3]. This surveillance formed part of the national crisis management plan organised in several phases. Phase 1 (from 2 January to 29 February 2020) involved the surveillance of individual cases and contact tracing to prevent the introduction of Sars-Cov-2 into the French territory. Phase 2 (from 1 to 13 March 2020) aimed to identify and break the chains of contamination to delay population transmission. Finally, phase 3 (since 14 March 2020) has involved population surveillance to reduce the dissemination of the virus within the population and mitigate its impact on the health care system [4]. From week 12 (16–22 March) to week 19 (4–10 May), a national lockdown was declared by the French government. At this time, health authorities advised the population to stay at home in the case of non-serious symptoms. However, patients were still allowed to go to a health care structure if they had an exemption certificate or call the emergency medical services (SAMU) or SOS Médecins.

Population surveillance was the cornerstone of crisis management by the French health authorities with varying objectives: monitoring the epidemiological dynamics at national and regional levels; identifying possible clusters; evaluating the impact of preventive measures (self-isolation, social distancing); evaluating the impact of the epidemic on the health of different populations (risk factors, vulnerable populations, immunity); and supporting the decisions of public health stakeholders.

This epidemiological surveillance relied on a range of different systems: existing systems such as the syndromic surveillance (SyS) system known as SurSaUD® (*Surveillance sanitaire des urgences et décès*) or the sentinel network of general practitioners (GPs); existing systems adapted for other purposes at the national or international level such as the information system for victim follow-up or the WHO outbreak investigation tool; and new systems set up during the COVID-19 crisis for patients in long-term care facilities, testing information, contact tracing, and cluster monitoring [5].

Developed in 2004 after the deadly 2003 heatwave, SurSaUD® collects daily data of individual visits to both emergency departments (EDs) and the network of emergency GPs known as SOS Médecins (SOSMed). The system also collects mortality data from the civil status offices and electronic death certificates [6, 7]. This non-specific surveillance aims to detect unexpected health events early on, monitor seasonal outbreaks, and assess outbreaks and their public health impact on the population [6, 8, 9]. SyS was already proven to adequately monitor

novel health emergencies [10, 11]. SyS was used from February 2020 at the start of the spread of COVID-19 across the entire French territory.

This study describes the characteristics of COVID-19-related visits in EDs and SOSMed associations at the national and regional levels from 17 February to 28 June 2020, a period that included the first nationwide lockdown. We also focused on the first days of use of SyS to monitor this exceptional event, especially in terms of its reactivity, its design and implementation, and any interaction with doctors. This could help other countries to implement a similar surveillance system for various emerging health situations of concern.

## Methods

### Materials

**Data from emergency departments.**   Individual data are collected daily from computerised medical records completed during ED consultations in the OSCOUR® (*Organisation de la surveillance coordonnée des urgences*) network, which grew from 23 EDs in 2004 to around 700 in 2020. This system records 93.3% of all ED visits in France, varying from 85.6% to 100% depending on the region, including the French overseas territories (except for Martinique). On average, 56,700 ED visits were recorded each day in 2019. Every morning, EDs transfer individual data from the previous 7 days to SpFrance. Most data (90%) are transmitted within 24 hours and consolidated within 72 hours.

For each visit, demographic data (birth date, gender, post code of residence), administrative data (date and time of admission and discharge, mode of transport, origin, destination after ED visit [hospitalisation, return home]), chief complaint in free text, and medical diagnoses are collected. Medical diagnoses correspond to the clinical information with a primary diagnosis (PD) and up to 10 secondary diagnoses (SD). They are coded using the International Classification of Diseases, version 10 (ICD-10). In 2019, PD was indicated in 77.5% of visits on average.

**Data from SOS Médecins.**   SOS Médecins is a network of emergency GP services providing emergency care in the private sector 24 hours a day and 7 days a week. They operate with hotlines that receive calls from patients, leading to the provision of medical advice, a home visit, or a consultation with a GP in a local SOSMed association.

Since 2006, the system has collected the data of individual visits on a daily basis. In 2020, 62 out of 63 SOSMed associations, mainly located in urban areas, participated in the SyS. All mainland regions have at least one SOSMed association in addition to Martinique. On average, 10,200 daily visits were recorded in 2019.

Every morning, SOSMed transfers individual data from the previous 3 days to SpFrance. Almost 97% of data are transmitted within 24 hours and consolidated within 72 hours. For each visit, demographic data (age, gender, post code of residence), administrative data (date, time, and origin of call), clinical information (using specific terms for diagnoses and chief complaints), and hospitalisation status are collected. Each visit can have up to three diagnoses. Among them, 95% have at least one medical diagnosis coded using a specific thesaurus.

Neither network attributes a unique identification number to patients. Since the goal of SyS is to measure the use of the health care system, surveillance is based on the number of visits instead of the monitoring of individual patients. Repeated visits of the same patient on distinctive days are counted separately.

### COVID-19-related visits

ED and SOSMed visits with suspected COVID-19 were identified based on accurate diagnosis codes that were jointly determined with field partners in the two networks.

**Table 1. ICD-10 codes of medical diagnosis used to identify emergency departments visits for suspected COVID-19.**

| ICD-10 codes | Labels | Date of recommendation by ATIH | Date of implementation in SurSAUD® |
|---|---|---|---|
| B34.2 | Coronavirus infection, unspecified site | - | Before February 2020 |
| B97.2 | Coronavirus as the cause of diseases classified elsewhere | - | Before February 2020 |
| U04.9 | Severe acute respiratory syndrome [SARS], unspecified | - | Before February 2020 |
| U07.1 | COVID-19, virus identified | 30 January 2020 | 24 February 2020 |
| U07.10 | COVID-19, respiratory symptoms, virus identified | 17 March 2020 | 19 March 2020 |
| U07.11 | COVID-19, respiratory presentation, virus not identified | 17 March 2020 | 19 March 2020 |
| U07.12 | Asymptomatic or symptomatic pauci-CoV-2 SARS carrier, virus identified | 17 March 2020 | 19 March 2020 |
| U07.14 | COVID-19, other clinical presentations, virus identified | 10 April 2020 | 4 May 2020 |
| U07.15 | COVID-19, other clinical presentations, virus not identified | 10 April 2020 | 4 May 2020 |

For ED visits, a new ICD-10 code (U07.1) based on WHO coding recommendations for COVID-19 was added to existing codes in the software (B34.2, B97.2, and U04.9) (Table 1) [12]. This list was enriched with extended codes created in March (U07.10, U07.11, and U07.12) and May 2020 (U07.14 and U07.15) by the French Agency for Information on Hospital Care (ATIH).

For SOSMed, visits were identified using the medical diagnosis code U07.1, which was introduced in early March 2020.

On 29 February (ED) and 3 March 2020 (SOSMed), coding recommendations were sent to all doctors, indicating that other diagnoses or symptoms such as cough, fever, respiratory failure, and dyspnoea should be coded as PD or SD in addition with one of the COVID-19-related codes.

COVID-19-related visits were categorised as those visits with at least one of the ICD-10 codes listed in Table 1 as either PD or SD.

## Analysis

The time course of the proportion of COVID-19-related visits among all coded visits in EDs and SOSMed were described according to age groups (all ages, under 15 years, 15–44 years, 45–64 years, 65 years and older) and gender from 17 February to 28 June 2020. Hospitalisations after visits were also monitored to assess the severity of patients' clinical conditions. The characteristics of COVID-19-related visits in EDs and SOSMed were also summarised for the lockdown period (16 March-10 May). Data analysis was stratified at both national and regional levels.

The distribution of the number of COVID-19-related visits in EDs by each ICD-10 code included in the case definition was analysed to assess which codes were effectively used by doctors.

Finally, the most frequent symptoms or pathologies associated with COVID-19-related visits in EDs (PD or SD) and SOSMed as well as the most frequent chief complaints for SOSMed visits were analysed.

The collected data formed part of the national surveillance system and did not include any identifiable personal information. Therefore, approval from an ethics committee was unnecessary.

## Results

### Outbreak description

**Overall study period.** During the study period, 634 ED and 60 SOSMed associations sent their data to SpFrance. At least one medical diagnosis was provided in 79.8% of ED visits and 95.2% of SOSMed visits.

From 17 February to 28 June, 170,113 COVID-19-related visits in EDs and 59,087 visits in SOSMed associations were recorded, corresponding to 4.0% and 5.7% of the total number of coded visits.

The first visits were recorded in February following the inclusion of the new diagnosis codes in EDs (17 February) and SOSMed (1 March). COVID-19-related visits were uncommon until March 8, representing less than 1.0% of overall daily visits.

From 9 March, visits sharply increased, reaching a peak by the end of the month. For all age groups, COVID-19-related visits peaked on 27 March 2020, reaching 22.6% (n = 5,853) and 25.6% (n = 1,777) of all ED and SOSMed visits, respectively (Fig 1). In April, COVID-19-related visits markedly and gradually declined. In June, the average daily proportion of COVID-19-related visits among overall visits stabilised at around 0.7% in EDs (n = 253) and 2.2% in SOSMed (n = 158) (Fig 1).

At the start of the lockdown period on 16 March, COVID-19-related visits represented 6.5% (n = 1,949) of overall daily visits in EDs and 10.8% (n = 1,127) in SOSMed. On 10 May, at the end of the lockdown, these visits had fallen to 3.5% (n = 999) and 5.8% (n = 311) of all ED and SOSMed visits, respectively.

The time course of COVID-19-related visits was concomitant across all age groups (Fig 1). The highest proportions of COVID-19-related visits were recorded among patients aged 45–64 years in the two networks and in the youngest adults (15–44 years) in SOSMed (Fig 1).

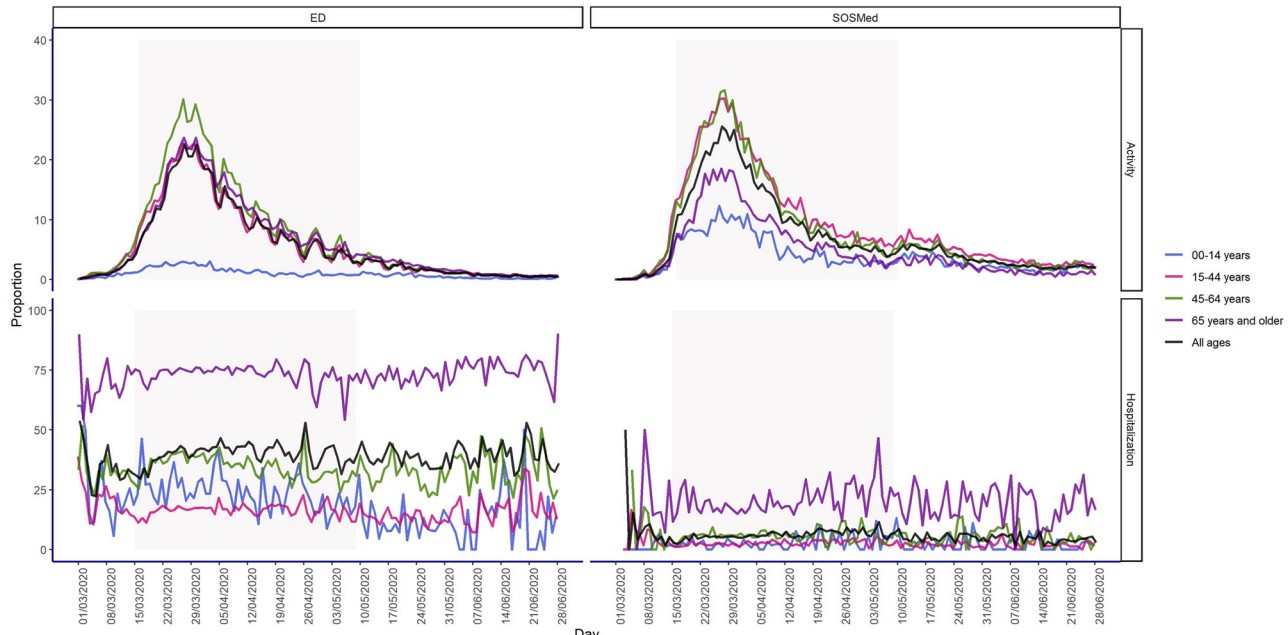

**Fig 1. Proportion of daily COVID-19-related visits among overall coded visits (top) and proportion of daily hospitalisations among overall COVID-19-related visits (bottom), by age group in emergency departments and SOS Médecins associations in France (including overseas territories and Corsica) from 17 February to 28 June 2020.**

During the peak, one out of three visits was related to COVID-19 in these age groups. Among the elderly (65 years and older) during the peak, 23.7% (n = 1,649) and 18.3% (n = 220) of ED and SOSMed visits, respectively, were linked to COVID-19. In SOSMed, COVID-19-related visits of children (under 15 years) showed a similar pattern to those of adults, reaching 12.1% of overall visits at the peak, whereas these visits in EDs were limited during the entire study period and reached 3% of overall visits in late March (Fig 1).

Non-COVID-19-related visits in EDs and SOSMed associations sharply decreased during the lockdown, reaching their lowest level 2 weeks after the beginning of the lockdown period. In early April 2020, a rebound in these visits was observed particularly in EDs (S1 Fig).

During the study period, 67,725 ED and 3,262 SOSMed patients were subsequently hospitalised after COVID-19-related visits, corresponding to 39.8% of all COVID-19-related visits in EDs and 5.5% in SOSMed associations. In EDs, this proportion ranged from 16.3% of 15-44-year-olds to 73.6% of patients aged 65 years and older, and in SOSMed, from 2.4% of children under 15 years to 19.3% of patients aged 65 years and older. In both networks, the proportion of COVID-19-related hospitalisations after visits remained relatively stable over the study period in all ages combined as well as in the different age groups and genders (Fig 1).

**Lockdown period.** The highest levels of daily activity in EDs and SOSMed associations were observed during the lockdown period. From 16 March to 10 May 2020, 140,011 ED and 46,038 SOSMed COVID-19-related visits were recorded, corresponding to 10.0% and 12.3% of overall visits. This respectively represented 82.3% and 77.9% of all COVID-19-related visits recorded in EDs and SOSMed during the study period.

In both networks, the majority of visits involved women: 54.3% in EDs and 58.1% in SOSMed (Table 2).

In SOSMed, 53.4% of visits were adults aged 15–44 years, whereas this age group corresponded to only 36.4% of ED visits. The population aged 45–64 years and 65 years and older were equally impacted in EDs (31.0% and 30.9%, respectively), whereas they were impacted differently in SOSMed, with the elderly being less represented (23.9% and 14.1%, respectively). Children under 15 years were the least affected, although they visited SOSMed associations (8.4%) more frequently than EDs (2.1%) (Table 2).

Almost 84.0% and 81.0% of hospitalisations following ED and SOSMed visits, respectively, were recorded during the lockdown period. Male patients were hospitalised more often than females after ED visits (47.5%) and SOSMed visits (6.7%), although women accounted for more than half of all hospitalisations after SOSMed visits (n = 1,365) (Table 2). Overall, more than half of all hospitalisations occurred in patients aged 65 years and older, with a higher proportion after ED (56.3%) compared to SOSMed visits (47.6%) (Table 2).

Among hospitalisations after ED visits, 1,967 patients were admitted to intensive care (3.5%). More than two-thirds were male (68.1%), and half aged 65 years and older (50.9%) (Table 2).

## Geographic pattern of COVID-19-related visits

The time course of the proportion of COVID-19-related visits among overall visits in EDs and SOSMed was similar in all mainland regions, and the peaks were reached simultaneously (Fig 2). In the French overseas territories, the time course of the pandemic was similar in Martinique but started later in Mayotte (April) and Reunion Island (mid-June).

For the entire study period, the highest proportions of COVID-19-related visits among overall visits in EDs and SOSMed associations were recorded in Corsica, Ile-de-France, Bourgogne-Franche-Comté, and Grand-Est (Table 3). The geographic distribution was more

**Table 2. Overall and COVID-19-related visits in emergency departments (EDs) and SOS Médecins associations by gender and age group during the lockdown period (16 March to 10 May 2020), France.**

| | Overall visits[a] (N) | COVID-19-related visits (N) | Proportion of COVID-19-related visits among overall visits (%) | Distribution of COVID-19-related visits (%) | Hospitalisations after visits (N) | Proportion of hospitalisations among COVID-19-related visits (%) | Hospitalisations in intensive care related to COVID-19 (N) | Proportion of hospitalisations in intensive care among COVID-19-related hospitalisation (%) |
|---|---|---|---|---|---|---|---|---|
| All ED visits | 1,395,573 | 140,011 | 10.0 | - | 56,696 | 40.5 | 1,967 | 3.5 |
| Gender | | | | | | | | |
| Missing | 197 | 29 | 14.7 | 0.0 | 7 | 24.1 | 1 | 14.3 |
| Female | 679,889 | 76,080 | 11.2 | 54.3 | 26,324 | 34.6 | 626 | 2.4 |
| Male | 715,487 | 63,902 | 8.9 | 45.6 | 30 365 | 47.5 | 1,340 | 4.4 |
| Age group (years) | | | | | | | | |
| Missing | 50 | 4 | 8.0 | 0.0 | 0 | 0 | NA | NA |
| Under 15 | 200,652 | 2,918 | 1.5 | 2.1 | 719 | 24.6 | 11 | 1.5 |
| 15–44 | 485,537 | 50,404 | 10.4 | 36.0 | 8,361 | 16.6 | 189 | 2.3 |
| 45–64 | 327,936 | 43,399 | 13.2 | 31.0 | 15,683 | 36.1 | 765 | 4.9 |
| 65 and older | 381,398 | 43,286 | 11.3 | 30.9 | 31,933 | 73.8 | 1,002 | 3.1 |
| All SOS Médecins visits | 373,167 | 46,038 | 12.3 | - | 2,649 | 5.8 | NA | NA |
| Gender | | | | | | | | |
| Missing | 333 | 49 | 14.7 | 0.1 | 1 | 2.0 | NA | NA |
| Female | 216,015 | 26,771 | 12.4 | 58.1 | 1,365 | 5.1 | NA | NA |
| Male | 156,819 | 19,218 | 12.3 | 41.7 | 1,283 | 6.7 | NA | NA |
| Age group (years) | | | | | | | | |
| Missing | 856 | 86 | 10.0 | 0.2 | 2 | 2.3 | NA | NA |
| Under 15 | 64,802 | 3,875 | 6.0 | 8.4 | 104 | 2.7 | NA | NA |
| 15–44 | 155,118 | 24,590 | 15.9 | 53.4 | 617 | 2.5 | NA | NA |
| 45–64 | 74,300 | 11,004 | 14.8 | 23.9 | 665 | 6.0 | NA | NA |
| 65 and older | 78,091 | 6,483 | 8.3 | 14.1 | 1,261 | 19.5 | NA | NA |

[a]: overall visits: visits with at least one coded medical diagnosis.

NA: not available.

heterogeneous at the district level (which is an administrative geographic level in France). Adults living in districts located in northeast France were the most impacted (Fig 3).

## Clinical characteristics of patients

**Distribution of ICD-10 codes used to identify COVID-19-related visits in emergency departments.** The nine ICD-10 codes used to identify COVID-19-related visits in EDs were used 171,185 times during the study period.

The code U07.1 ("coronavirus disease") was used by doctors immediately after its inclusion in ED software in mid-February. It was also the most frequently used code (42.3%, n = 72,411) (Fig 4).

The other frequently used codes were U07.11 ("COVID-19, clinical case, virus not identified"; 32.8%, n = 56,228), U07.10 ("COVID-19, virus identified"; 7.5%, n = 12,756), B34.2

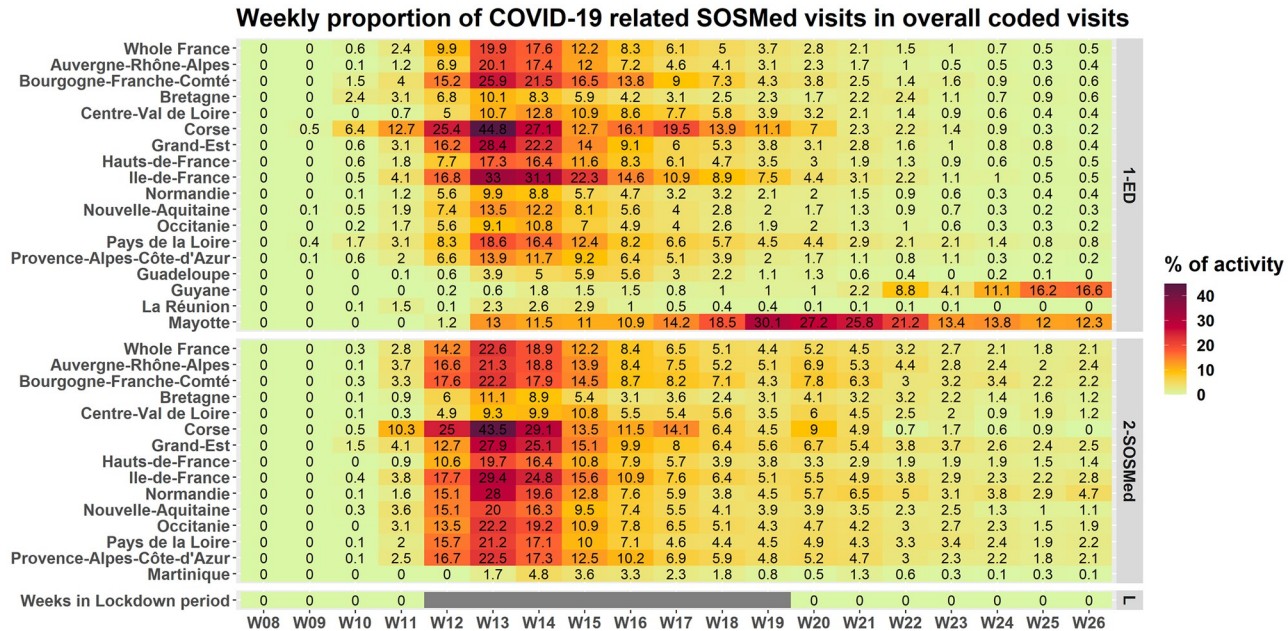

**Fig 2. Proportion of weekly COVID-19-related visits among overall visits in emergency departments and SOS Médecins associations, all ages, from 17 February to 28 June 2020, in the French regions and overseas territories.**

("Coronavirus infection, unspecified site"; 7.4%, n = 12,593), and B97.2 ("Coronavirus as the cause of diseases classified elsewhere"; 6.4%, n = 11,013). The code U07.15 ("COVID-19 non-respiratory form, virus not identified"), which was introduced into the case definition in May, was used in 2.0% of visits (n = 3,481), while U07.14 ("COVID-19 non-respiratory form, virus identified") and U04.9 ("severe acute respiratory syndrome (SARS), unspecified") were rarely employed (Fig 4).

In early March, U07.1 accounted for 72% of codes used by doctors, although its use progressively declined to 28.0% by late June. By contrast, the use of U07.10 and U07.11 gradually increased from mid-March and respectively accounted for 10.6% and 32.8% of codes used by late June. The proportion of B34.2 and B97.2 codes also decreased and stabilised at around 4% to 6% of all codes by late June. These codes were used by 510 EDs in France. In these EDs, the new ICD-10 codes were also employed during the study period, meaning that the use of B34.2 and B97.2 was not related to technical issues concerning software updates.

When U07.15 was introduced in May, this code rapidly represented 11% of all COVID-19-related codes and continued to slowly increase, reaching 15.9% of total uses by late June.

**Other diseases associated with COVID-19 diagnosis.** Medical codes for suspected COVID-19 were mainly used as the PD (N = 133,196 ED visits; 77.9% of all COVID-19-related visits). Among these visits, 14,265 (10.7%) had an average of 1.3 SDs. A wide range of diagnoses were reported, with the most common being pulmonary infectious disease (pneumonia, bronchitis, bronchiolitis) (10.9%), heart disease (high blood pressure, cardiac failure, pulmonary embolism) (8.1%), COVID-19 (7.7%), dyspnoea (6.6%), and cough (4.9%) (S1 Table).

In SOSMed associations, suspected COVID-19 was recorded as the first diagnosis in 47,718 visits (81%). Among these visits, 1,761 visits (3.7%) had an average of 1.1 associated diagnoses. As in EDs, associated diagnoses were infrequent, with the most common being ENT diseases (rhinopharyngitis, angina) (24.1%), gastroenteritis (8.9%), acute bronchitis (7.3%), acute pneumonia (6.6%), anxiety (5.9%), and influenza-like illness (S2 Table).

**Table 3. Overall and COVID-19-related visits in emergency departments and SOS Médecins associations by region during the lockdown period (16 March to 10 May 2020), France.**

| | Emergency departments | | | | | SOS Médecins | | | | |
|---|---|---|---|---|---|---|---|---|---|---|
| | Overall visits[a] (N) | COVID-19-related visits (N) | Proportion of COVID-19-related visits among overall visits (%) | Hospitalisations after visits (N) | Proportion of hospitalisations following COVID-19-related visits (%) | Overall visits[a] (N) | COVID-19-related visits (N) | Proportion of COVID-19-related visit among overall visits (%) | Hospitalisations after visits (N) | Proportion of hospitalisations following COVID-19-related visits (%) |
| **Mainland regions** | | | | | | | | | | |
| Auvergne-Rhône-Alpes | 159,937 | 14,421 | 9.0 | 6,209 | 43.1 | 42,414 | 5,398 | 12.5 | 401 | 7.4 |
| Bourgogne-Franche-Comté | 67,412 | 9,298 | 13.8 | 3,850 | 41.4 | 10,988 | 1,527 | 12.9 | 58 | 3.8 |
| Brittany | 68,474 | 3,587 | 5.2 | 1,702 | 47.4 | 16,259 | 979 | 5.6 | 84 | 8.6 |
| Centre-Val de Loire | 54,324 | 4,309 | 7.9 | 1,100 | 25.5 | 12,398 | 888 | 6.7 | 23 | 2.6 |
| Corsica | 7,988 | 1,692 | 21.2 | 1,632 | 96.5 | 1,936 | 381 | 19.7 | 0 | 0.0 |
| Grand-Est | 112,827 | 14,701 | 13.0 | 7,330 | 49.9 | 34,457 | 5,168 | 14.5 | 383 | 7.4 |
| Hauts-de-France | 114,233 | 10,350 | 9.1 | 3,786 | 36.6 | 32,374 | 3,493 | 10.3 | 126 | 3.6 |
| Ile-de-France | 248,166 | 45,815 | 18.5 | 14,500 | 31.6 | 65,214 | 10,486 | 15.8 | 668 | 6.4 |
| Normandy | 72,176 | 3,747 | 5.2 | 1,471 | 39.3 | 18,070 | 2,264 | 12.5 | 113 | 5.0 |
| Nouvelle-Aquitaine | 129,312 | 8,505 | 6.6 | 3,842 | 45.2 | 52,732 | 5,682 | 10.7 | 303 | 5.3 |
| Occitanie | 127,082 | 6,888 | 5.4 | 3,074 | 44.6 | 19,297 | 2,242 | 11.4 | 76 | 3.4 |
| Pays de la Loire | 62,589 | 6,106 | 9.8 | 3,100 | 50.8 | 22,116 | 2,470 | 10.9 | 159 | 6.4 |
| Provence-Alpes-Côte d'Azur | 129,539 | 9,191 | 7.1 | 4,689 | 51.0 | 39,583 | 4,930 | 12.5 | 245 | 5.0 |
| **Overseas territories** | | | | | | | | | | |
| Guadeloupe | 10,059 | 328 | 3.3 | 116 | 35.4 | NA | NA | NA | NA | NA |
| Martinique | NA | NA | NA | NA | NA | 5,329 | 130 | 2.2 | 10 | 7.7 |
| French Guiana | 8,544 | 87 | 1.0 | 26 | 29.9 | NA | NA | NA | NA | NA |
| Reunion Island | 17,226 | 206 | 1.2 | 128 | 62.1 | NA | NA | NA | NA | NA |
| Mayotte | 5,339 | 779 | 14.6 | 140 | 18.0 | NA | NA | NA | NA | NA |

[a]: overall visits: visits with at least one coded medical diagnosis.

NA: not available.

The most common complaints were fever and sweat (18.0%), cough (16.0%), ENT diseases (sore throat, cold) (11.3%), gastrointestinal disorders (diarrhoea, vomiting, abdominal pain) (9.4%), and headaches (8.8%) (S3 Table).

## Discussion

From 17 February to 28 June, 170,113 ED visits and 59,087 SOSMed visits were related to COVID-19 in France. These visits corresponded to a higher proportion of overall visits in SOSMed (5.6%) than in EDs (4.0%). The majority were recorded during the national lockdown period. The time course of the visits followed a similar trend in both networks, in the different age groups, and in all regions of mainland France. The visits mostly involved women and younger patients (frequently aged 15–44 years). Children were less concerned, although they visited SOSMed associations more often than EDs. Hospitalisations after visits were predominantly observed in men and the elderly (65 years and older). Fever and sweat, dyspnoea, cough, pneumonia, and influenza-like illness were the most common diagnoses associated with COVID-19-related visits. Districts located in northeast France were the most impacted by this epidemic wave.

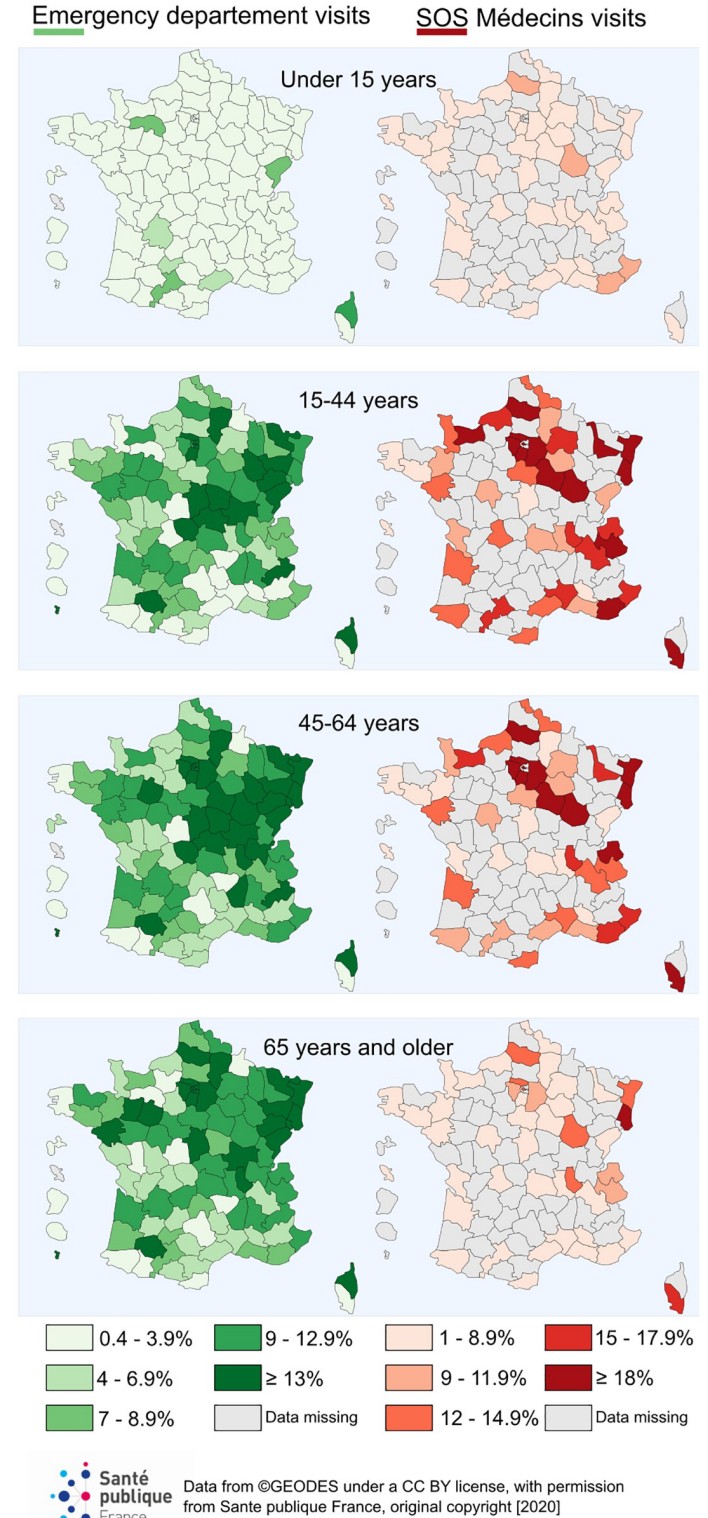

**Fig 3. Proportion of COVID-19-related visits among overall visits in emergency departments and SOS Médecins associations by districts and age group during the lockdown period (from 16 March to 10 May 2020) in France.**

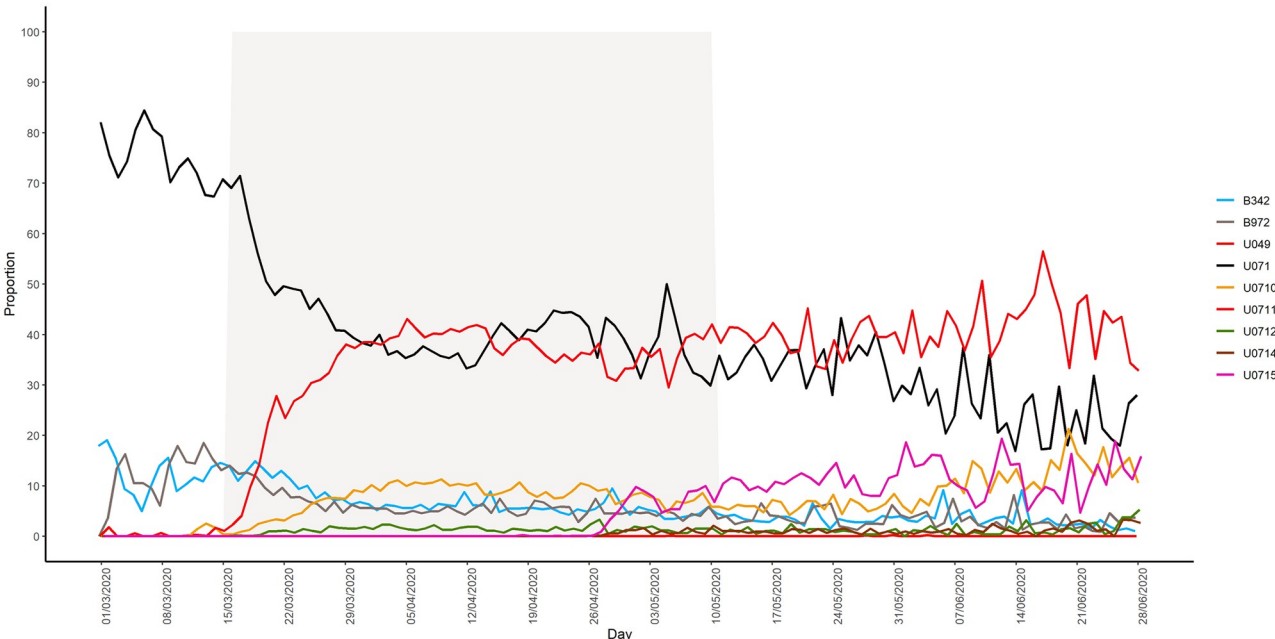

**Fig 4. Daily proportion of the number of emergency departments (ED) visits for each of the ICD-10 codes included in the definition of the COVID-19 indicator among the total number of COVID-19-related visits in EDs from 1 March to 28 June 2020.**

## Results consistent with other data sources

Our observations are consistent with other data sources used for monitoring the outbreak in France (laboratory-confirmed tests, hospital and intensive care admissions for COVID-19, COVID-19-related deaths) [5, 13]. The population usually visiting SOSMed (younger than in EDs) and the urban location of these associations could partly explain the higher rate of children attending COVID-19-related visits in SOSMed. As reported in northeast France as well as in international studies, male gender and advanced age were both related to severe COVID-19 and death [14–17]. COVID-19 surveillance in the US showed that 75.5% of recorded cases were aged 18–64 years, and 52.1% were women, whereas among hospitalised patients with a confirmed laboratory test, 42.5% were aged 65 years and older, and 50.6% were men [18]. Regarding severity in men, as suggested in several publications, it could be explained by different factors such as the immune system, sex hormones, physiological factors, sociocultural factors affecting health, and underlying comorbidities [17, 19, 20]. In our study, associated clinical symptoms or diagnoses were known to be strongly correlated to COVID-19 cases [21, 22]. Early in the epidemic, these indicators were monitored as proxies due to the lack of specific ICD-10 codes and the scarcity of PCR tests to accurately identify visits related to this emergent disease.

Disparities in the geographic distribution of COVID-19-related visits were mainly explained by the spatial spread of the epidemic, particularly in districts where many clusters were registered during the outbreak. Further studies will be required to examine other factors that may explain these disparities such as health care geography and the implementation of specific COVID-19 health care measures.

In the French overseas territories, the onset of the outbreak was later in Mayotte and French Guiana than in the mainland regions, which is in accordance with the delayed spread of the epidemic in these remote territories.

## Identification of COVID-19-related visits

In the two networks, patients suffering from COVID-19 were identified based on a clinical examination, since biological tests for SARS-COV2 infection were not yet available at the beginning of the study period. RT-PCR tests were progressively introduced from May 2020, although the results were poorly recorded in the system. This data collection was improved during the second wave of the epidemic following the development of rapid biological tests. However, doctors' ability to identify patients with COVID-19 in EDs and SOSMed associations may have improved with their better scientific knowledge of the disease. One example is the identification of dysgeusia as a symptom of this disease.

We specifically monitored COVID-19-related visits by considering the set of ICD-10 codes listed in Table 1. A subset of patients may have only been diagnosed with proxy-indicators (cough, fever, respiratory failure, and dyspnea) rather than one of the ICD-10 codes of interest, particularly in the early days of the epidemic. However, we rapidly gave all EDs and SOSMed associations specific coding recommendations for COVID-19-related visits as well as for visits with proxy-symptoms and diseases in order to reduce this risk.

Specific COVID-19 health care measures may have been implemented at the local level during the crisis in March and April to separate patients with or without COVID-19 symptoms. However, data relating to these measures were not systematically entered into the system. The number of ED visits may thus have been underestimated. On the contrary, EDs occasionally recorded COVID-19-related visits with the diagnosis U07.1, even though the reason for the visits was the need for biological testing, because the patients (without clinical symptoms) were contacts of a positive COVID-19 case. The correct code for these visits was rather U07.13 (which was excluded from our case definition). For example, this was the case in EDs in the districts of Haute-Corse, Morbihan, and Cher. When these miscoded visits were identified, it was requested for the wrong codes to be corrected where possible. Despite the possible bias in the identification of COVID-19-related visits, we assume that the coding recommendations given to all EDs and SOSMed associations by the heads of these two networks at the start of the surveillance period contributed to ensuring the consistent surveillance of COVID-19 across the French territory.

During the lockdown period, the use of health care services drastically decreased, partly because the population was fearful about being contaminated in hospitals or doctors' offices. Our surveillance system may have identified patients with symptoms and/or severe medical conditions, while asymptomatic patients or those without severe symptoms might not have consulted in EDs. This limitation would apply to SOSMed visits to a lesser extent, although it should be recalled that the SOSMed network only covers the most populated cities. The use of two complementary data sources nevertheless represents one of the strengths of this study, since they were able to capture patients with different health care behaviours.

## Strengths and role of the syndromic system for COVID-19 surveillance

The syndromic surveillance system known as SurSaUD® has already proven its benefit in rapidly detecting unusual variations in the number of ED and SOSMed visits, adequately monitoring changes in the variation of seasonal or unexpected outbreaks, and contributing to the impact assessment of events on the population [9, 11]. Based on the automatic daily collection of individual data, the system was the first to monitor the spread of emergent pathologies such as COVID-19 in the population. The early implementation and use of new diagnosis codes in EDs and SOSMed associations also highlights the flexibility and adaptability of this system to a large extent. The rapid introduction of a new code was previously implemented in a limited territory to monitor the dengue outbreak in Reunion Island in 2018 [23].

The partnership with data providers (ED and SOSMed doctors as well as the Federation of the Regional Observatories of Emergencies) is one of the strengths of the SyS system. In addition to selecting and distributing the coding recommendations to all doctors, the feedback received from their field experience during the weekly meetings helped support the interpretation of patterns obtained from the data analysis. This qualitative analysis was fruitful during the early period of the emergent disease to suggest hypotheses about possible disease symptoms, understand the fear and behaviour of patients regarding their health care, or correctly interpret the epidemic curve.

Data analysis of SyS (with other data sources) formed part of the daily reports shared with decision-makers at the national and regional levels as well as the weekly national and regional bulletins published every Thursday on the SpFrance website [5]. COVID-19-related visits in EDs were also used in complement with other data sources for different objectives: estimating the reproduction number R of the epidemic and providing criteria to determine the end of the lockdown [13, 24]. SyS data were also used by hospitals to monitor their occupancy rates and manage their needs for intensive care beds [25].

This descriptive study provides a comprehensive picture of the COVID-19 outbreak in emergency health care settings. It also highlighted how syndromic surveillance can be adapted to rapidly monitor a new emergent virus and provide real-time information to health authorities for decision-making purposes.

In addition to COVID-19 surveillance, the ED and SOSMed data are unique data sources that can monitor real-time visits for other common diseases (infectious diseases, cardiovascular diseases, and mental illness) and characterise the impact of the lockdown period on health care more broadly.

After a summer period marked by fewer COVID-19-related visits in the two networks, the second wave began in September 2020, with a sharp increase of visits in October. To control this second wave, national and local authorities implemented a series of mitigation measures at the local and national levels, including a second nationwide lockdown from 30 October to 15 December 2020 along with curfews [26]. In complement to the other sources (laboratory tests, long-term care facilities, intensive care, etc.), EDs and SOSMed associations were still used to monitor the epidemic during this second wave and beyond. Future studies will be conducted to compare the characteristics of COVID-19-related visits during the different waves.

## Supporting information

**S1 Fig. Number of COVID-19 and non-COVID-19-related visits in emergency departments and SOS Médecins associations from 17 February to 28 June 2020.**
(TIF)

**S1 Table. Diagnoses associated with COVID-19-related emergency department visits from 17 February to 28 June 2020.**
(DOCX)

**S2 Table. Diagnoses associated with COVID-19-related SOSMed visits from 17 February to 28 June 2020.**
(DOCX)

**S3 Table. Most common complaints reported in COVID-19-related SOSMed visits from 17 February to 28 June 2020.**
(DOCX)

## Acknowledgments

The authors acknowledge the contribution made by the emergency departments and clinicians involved in the OSCOUR and SOS Médecins networks in France; the ongoing support of the Federation of the Regional Observatories of Emergencies; the Scientific Society of Emergency Medicine; and the Federation of Emergency General Practitioners' (SOS Médecins) associations. We also thank Nicolas Vincent (Santé publique France Centre-Val-de-Loire, Orléans, France), Erica Fougère (Santé publique France Auvergne-Rhône-Alpes, Lyon, France), Laure Meurice (Santé publique France Nouvelle-Aquitaine, Bordeaux, France), Noémie Fortin (Santé publique France Pays-de-la-Loire, Nantes, France), Pierre-Henry Juan (SOS Médecins, Lyon, France), Agnès Barondeau-Leuret (Fédération des Observatoires régionaux des urgences (FEDORU), Bourgogne-Franche-Comté, France), and Karim Tazarourte (Hospices civils de Lyon, Lyon, France) for reviewing the manuscript.

We also thank the following co-author groups:

**SurSaUD® Regional Focal Point**: Nicolas Vincent (Nicolas.VINCENT@santepublique-france.fr, Santé publique France Centre-Val-de-Loire, Orléans, France), Audrey Andrieu (Santé publique France Antilles, Cayenne, Guyane), Arnoo Shaiykova (Santé publique France Hauts-de-France, Lille, France), Nahida Atiki (Santé publique France Normandie, Rouen, France), Oriane Broustal (Santé publique France Grand-Est, Nancy, France), Delphine Casamatta (Santé publique France Auvergne-Rhône-Alpes, Lyon, France), Sonia Chêne (Santé publique France Bourgogne-Franche-Comté, Dijon, France), Jamel Daoudi (Santé publique France Océan Indien, Saint-Denis, La Réunion), Elise Daudens-Vaysse (Santé publique France Antilles, Fort-de-France, Martinique), Joël Deniau (Santé publique France Provence Alpes Côte d'Azur-Corse, Marseille, France), Marlène Faisant (Santé publique France Bretagne, Rennes, France), Noemie Fortin (Santé publique France Pays-de-la-Loire, Nantes, France), Erica Fougère (Santé publique France Auvergne-Rhône-Alpes, Lyon, France), Celine Francois (Santé publique France Ile-de-France, Saint-Denis, France), Laure Meurice (Santé publique France Nouvelle-Aquitaine, Bordeaux, France), Jérôme Pouey (Santé publique France Occitanie, Toulouse, France), and Leslie Simac (Santé publique France Occitanie, Montpellier, France).

**SOS Médecins France National Board**: Pierre-Henry Juan (pierre.henry.juan@gmail.com, SOS Médecins Lyon), Jean-Christophe Masseron (SOS Médecins Chambéry), Serge Smadja (SOS Médecins Paris), Pascal Chansard (SOS Médecins Paris), and Patrick Guérin (SOS Médecins Nantes).

**Group of Emergency Medicine Doctors**: Agnès Barondeau-Leuret (agnes.leuret@rubfc.fr; Observatoire régional des urgences Bourgogne-Franche-Comté, France), Mohamed Hachelaf (Centre Hospitalier de Besançon, France), Thibault Desmettre (Centre Hospitalier Universitaire de Besançon, France), Pierre-Yves Gueugniaud (SAMU Lyon, France), Abdeslam Redjaline (Observatoire régional des urgences Auvergne-Rhône-Alpes, France), Jeannot Schmidt (Centre Hospitalier Universitaire de Clermont-Ferrand, France), Claude Zamour (Centre Hospitalier de Valence, France), Maurice Raphael (Centre Hospitalier Universitaire Kremlin-Bicêtre, France), Christophe Leroy (Assistance Publique—Hôpitaux de Paris, France), Laurent Maillard (Centre Hospitalier d'Agen, France), Sandrine Charpentier (Centre Hospitalier Universitaire de Toulouse, France), Andre de Caffarelli (Centre Hospitalier de Bastia, France), Gilles Viudes (Observatoire régional des urgences Provence-Alpes-Côte d'Azur, France), Philippe Garitaine (Centre Hospitalier de Saint-Tropez, France), Vincent Pommier de Santi (Centre d'épidémiologie et de santé publique des armées, Marseille, France), Bruno Maire (Est-Rescue/Réseau and Observatoire Urgences Grand-Est, France), Marc Noizet (Centre hospitalier de Mulhouse, France), Patrick Miroux (Centre Hospitalier Universitaire d'Angers,

France), Marie-Astrid Metten (Observatoire régional des urgences Pays de la Loire, France), Jean-François Buyck (Observatoire régional de la santé des Pays de la Loire, France), Mélanie Goument (Centre Hospitalier de Nantes, France), Christèle Gras-Le Guen (Centre Hospitalier de Nantes, France), Françoise Cellier (Centre Hospitalier d'Yves Le Foll, Saint-Brieuc, France), Pierre Kergaravat (Centre Hospitalier d'Yves Le Foll, Saint-Brieuc, France), Hervé Mourou (Observatoire régional des urgences Occitanie, France), Patrick Mauriaucourt (Centre Hospitalier Universitaire de Lille, France), and Philippe Linassier (Centre Hospitalier Régional d'Orléans, France).

## Author Contributions

**Conceptualization:** Marie-Michèle Thiam, Gaëlle Pedrono, Anne Fouillet, Céline Caserio-Schönemann.

**Data curation:** Marie-Michèle Thiam, Isabelle Pontais, Cécile Forgeot, Anne Fouillet.

**Formal analysis:** Marie-Michèle Thiam, Isabelle Pontais, Cécile Forgeot, Gaëlle Pedrono, Louis-Marie Paget, Anne Fouillet.

**Investigation:** Marie-Michèle Thiam, Isabelle Pontais, Cécile Forgeot, Louis-Marie Paget, Anne Fouillet.

**Methodology:** Marie-Michèle Thiam, Isabelle Pontais, Cécile Forgeot, Gaëlle Pedrono, Louis-Marie Paget, Anne Fouillet, Céline Caserio-Schönemann.

**Project administration:** Marie-Michèle Thiam, Anne Fouillet, Céline Caserio-Schönemann.

**Software:** Isabelle Pontais, Cécile Forgeot.

**Supervision:** Anne Fouillet, Céline Caserio-Schönemann.

**Validation:** Marie-Michèle Thiam, Isabelle Pontais, Cécile Forgeot, Gaëlle Pedrono, Louis-Marie Paget, Anne Fouillet, Céline Caserio-Schönemann.

**Writing – original draft:** Marie-Michèle Thiam, Anne Fouillet.

**Writing – review & editing:** Marie-Michèle Thiam, Isabelle Pontais, Cécile Forgeot, Gaëlle Pedrono, Louis-Marie Paget, Anne Fouillet, Céline Caserio-Schönemann.

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
