## [Decision Letter · Decision Letter 0]

4 Jul 2021

PONE-D-21-12135

Syndromic surveillance: A key component of community-based health monitoring during the first wave of the COVID-19 outbreak in France, February-June 2020

PLOS ONE

Dear Dr. Thiam,

Thank you for submitting your manuscript to PLOS ONE. After careful consideration, we feel that it has merit but does not fully meet PLOS ONE’s publication criteria as it currently stands. Therefore, we invite you to submit a revised version of the manuscript that addresses the points raised during the review process.

Both reviewers agree that the study is timely and important. However, Reviewer 1 felt that more details on data collected on the hospital visits should be provided. I share this same opinion with Reviewer 1 that such details would improve the manuscript, and make for a compelling read.

We look forward to receiving your revised manuscript.

Kind regards,

Siew Ann Cheong, Ph.D.

Academic Editor

PLOS ONE

Journal Requirements:

2. To comply with PLOS ONE submission guidelines, in your Methods section, please provide additional information regarding your statistical analyses. For more information on PLOS ONE's expectations for statistical reporting, please see https://journals.plos.org/plosone/s/submission-guidelines.#loc-statistical-reporting.

3. Please list the name and version of any software package used for statistical analysis, alongside any relevant references. For more information on PLOS ONE's expectations for statistical reporting, please see https://journals.plos.org/plosone/s/submission-guidelines.#loc-statistical-reporting.

4. One of the noted authors is a group or consortium [SurSaUD® Regional Focal Point, SOS Médecins, Group of Emergency Medicine Doctors]. In addition to naming the author group, please list the individual authors and affiliations within this group in the acknowledgments section of your manuscript. Please also indicate clearly a lead author for this group along with a contact email address.

5. Please amend your list of authors on the manuscript to ensure that each author is linked to an affiliation. Authors’ affiliations should reflect the institution where the work was done (if authors moved subsequently, you can also list the new affiliation stating “current affiliation:….” as necessary).

6. We note that Figure 3 and Table S1 in your submission contain map images which may be copyrighted. All PLOS content is published under the Creative Commons Attribution License (CC BY 4.0), which means that the manuscript, images, and Supporting Information files will be freely available online, and any third party is permitted to access, download, copy, distribute, and use these materials in any way, even commercially, with proper attribution. For these reasons, we cannot publish previously copyrighted maps or satellite images created using proprietary data, such as Google software (Google Maps, Street View, and Earth). For more information, see our copyright guidelines: http://journals.plos.org/plosone/s/licenses-and-copyright.

6.1.    You may seek permission from the original copyright holder of Figure 3 and Table S1 to publish the content specifically under the CC BY 4.0 license. 

6.2.    If you are unable to obtain permission from the original copyright holder to publish these figures under the CC BY 4.0 license or if the copyright holder’s requirements are incompatible with the CC BY 4.0 license, please either i) remove the figure or ii) supply a replacement figure that complies with the CC BY 4.0 license. Please check copyright information on all replacement figures and update the figure caption with source information. If applicable, please specify in the figure caption text when a figure is similar but not identical to the original image and is therefore for illustrative purposes only.

7. Please include your tables as part of your main manuscript and remove the individual files. Please note that supplementary tables should remain uploaded) as separate "supporting information" files.

Reviewers' comments:

Reviewer's Responses to Questions

**Comments to the Author**

1. Is the manuscript technically sound, and do the data support the conclusions?

Reviewer #1: Yes

Reviewer #2: Yes

2. Has the statistical analysis been performed appropriately and rigorously? 

Reviewer #1: Yes

Reviewer #2: Yes

3. Have the authors made all data underlying the findings in their manuscript fully available?

Reviewer #1: Yes

Reviewer #2: Yes

4. Is the manuscript presented in an intelligible fashion and written in standard English?

Reviewer #1: Yes

Reviewer #2: Yes

5. Review Comments to the Author

Reviewer #1: The manuscript presents a solid descriptive study and appraisal of the French SurSaUD surveillance system during the first wave of COVID-19 outbreak (February-June 2020). The data aggregates came from a widely covering network of both emergency departments (EDs) and emergency general practitioners (SOSMed). The COVID-19-related visits were identified using a government-recommended set of ICD-10 codes. Chronological trends were summarized, with respect to demographic, geographic, and clinical attributes of the cases. The results were overall very informative and demonstrated value of the surveillance system. The manuscript is well organized but could be improved in terms of rigor and clarity (see major comments).

Major comments:

1. Please introduce more contexts about the French government policies that affected ED and SOSMed visits, especially during the lockdown. Did the patients have freedom to just walk in or need referral/approval during that special period?

2. The fact that no unique identifier in the systems may be of concern, especially if duplicative counts were prevalent. What was the estimated rate of duplicates in counting the visits?

3. Did the left and right panels of Figure 1 use different normalization approaches? In the left panel it seems each age group has its own denominator, while in the right panel it seems the suspected cases were used as the single shared denominator across the ages groups so that the “All age group” percentage appears to be the sum of the others. Please clarify or consider using a more consistent way of normalizing.

4. In Table 2 ED visits, please clarify why the column Distribution of COVID-19-related visits there is 0.0% Missing Gender, but in the column Proportion of hospitalisations following COVID-19-related visits there is 24.1% Missing Gender? How could that happen if the latter was a subset of the former?

5. How did the baseline data transmission volumes change during the study period? It would be informative to visualize by using stacked area charts (especially with raw counts) that show the non-COVID-related visits at the bottom and the COVID-related visits stacked on top over time.

6. Was there a possible explanation for the higher rate of female COVID-19-related visits? For example, it would be informative to break down by age group distribution to see if there was additional caution in caring for pregnant women.

7. What was the insurance mechanism for visiting SOSMed versus ED in France? Did the higher rate of children visiting SOSMed than ED indicate potential socioeconomic disparity? Please discuss.

Minor comments:

1. The notion of “community” is highlighted in the title and several places. However, there lacks any contrast example of “non-community” surveillance to help readers appreciate the uniqueness of the SurSaUD system.

2. If the COVID-19 code set was evolving along with the proficiency that the coders gradually gained over time, how could the trends be interpreted reliably with such confounding effects?

3. Were there any observed differences in the COVID-19 coding practice (e.g., distribution of common code usage) compared to other countries that also used ICD-10 (if published).

4. Please enhance the clarity when the word “department(s)” or “departmental” is mentioned by itself.

5. Please add a citation after mentioning of the reproduction number.

6. The last part of Discussion involves a good amount of promotion materials, which could be reduced to allow for strengthening points around the study results.

7. Since it has been a year, please briefly discuss whether (or not) the system still sustains and has been able to function consistently in surveillance of the later waves of the epidemic.

Reviewer #2: The authors report the epidemiological characteristics of those with a COVID-19-related visit captured by 2 health monitoring systems (emergency department visits and SOS Médecins) from February 17th to June 28th, 2020. Characteristics of this population during a national lockdown period between March 16th and May 10th were also described. COVID-19 related visits were categorized by any one of 9 specified COVID-19 specific ICD-10 codes. Results were also reported by geographic area in France. It was found that 4.0% of all ED visits and 5.6% of all SOSMed visits were COVID-19-related. The peak was observed in the nationwide lockdown period. The authors conclude that these syndromic surveillance systems allow for community-based health monitoring of COVID-19 in France.

This is a timely and relevant study that adds to the existing literature of syndromic surveillance tools in complementing laboratory based/PCR based surveillance of COVID-19. Figures and tables are clear. The authors adequately reported summary statistics and used appropriate methodologies with a few suggestions and some areas in need of improvement. Overall more can be added to highlight why this study is needed and how the results will be used.

Background: Details about each wave should be moved to a new first section in Methods which outlines key information about the study setting and context. Also for background section, please elaborate further on why this study needed to be done.

Discussion- there is mention that the results are “consistent with other data sources used for monitoring the outbreak in France”- please tell us more about these other sources, and what the data presented here adds to what is already known. It would also be important to outline how the trends reported here compared to confirmed COVID-19 laboratory cases and confirmed hospitalizations. Finally, the discussion lacks a concluding paragraph which clearly highlights what this study has contributed.

Specific suggestions:

1.) (Methods P4) “orientation after ED visit” should be changed to “destination after ED visit”

2.) (Methods P5) “All visits with a suspicion of COVID-19 recorded as PD or SD were selected for this study.” My understanding is visits that had one of the ICD-10 codes as either the PD or the SD were identified as a COVID-19-related visit. Please make this more clear and explicitly state that “COVID-19 related visits were categorized as those visits with at least 1 of the ICD-10 codes listed in table 1 as either the PD or the SD.”

3.) (Methods P5) “Other diagnoses or symptoms such as cough, fever, respiratory failure, and dyspnoea were be coded as PD or SD in addition with one of the COVID-19-related codes.” Is there a reason why the presence of certain symptoms (without a qualifying ICD code) was not used in your definition of COVID-19-related visit? It is possible that a subset of patients may have only receive a diagnosis of “cough” and not one of the ICD-10 codes listed in table 1. Thus, some possible COVID-19-related visits may have been missed. If symptoms cannot be added to the definition of “COVID-19-related visit”, then a limitations paragraph should be included which states that this study was only able to identify potential COVID-19 visits that were assigned an ICD-10 code by a healthcare provider.

- See https://bmcpublichealth.biomedcentral.com/articles/10.1186/s12889-021-11303-9

which uses SyS in Canada and uses syndrome which were predefined groups of symptoms from ED to identify potential COVID-19 Cases

4.) Be consistent with the lockdown period. In some areas of the manuscript, the March 16th - May 11th time period was used while in other the March 16th - May 10th time period was used.

5.) Methods: Was the data pulled on one date at the end of the study period or was it extracted weekly/daily throughout the study period? Please specify this in the methods. Due to data back tracking, data can change depending on when it was extracted.

6.) (Results P17) “As in EDs, associated diagnoses were infrequent, with the most common being ENT diseases (rhinopharyngitis, angina) (24.1%). Please clarify how angina is considered an ENT disease?

7.) (Discussion P20) “On the contrary, we occasionally observed that visits for biological testing were miscoded (coded as U07.1 instead of U07.13, which was not included in our case definition).” This sentence may inadvertently been reversed- suggest rephrasing this as “ we occasionally observed that visits for biological testing were miscoded: instead of U07.1 they were coded as U07.13, which was not included in our definition.”

6. PLOS authors have the option to publish the peer review history of their article (what does this mean?). If published, this will include your full peer review and any attached files.

Reviewer #1: No

Reviewer #2: **Yes: **Arjuna Maharaj

---

## [Author Response · Author response to Decision Letter 0]

25 Aug 2021

Answers to reviewers

We thank the reviewers for the time spent on this review and for the comments. We have taken into account all the recommendations. Detail of the answers to the comments are below (in bold). For some suggestions made by the reviewers, we added extract of the manuscript in italic and highlighted modifications in yellow.

A. Review Comments to the Author

Reviewer #1 (#1): The manuscript presents a solid descriptive study and appraisal of the French SurSaUD surveillance system during the first wave of COVID-19 outbreak (February-June 2020). The data aggregates came from a widely covering network of both emergency departments (EDs) and emergency general practitioners (SOSMed). The COVID-19-related visits were identified using a government-recommended set of ICD-10 codes. Chronological trends were summarized, with respect to demographic, geographic, and clinical attributes of the cases. The results were overall very informative and demonstrated value of the surveillance system. The manuscript is well organized but could be improved in terms of rigor and clarity (see major comments).

Thank you for your appreciation

Major comments:

Comment 1#1.Please introduce more contexts about the French government policies that affected ED and SOSMed visits, especially during the lockdown. Did the patients have freedom to just walk in or need referral/approval during that special period?

In France, at the beginning of the first lockdown, the government recommended to the population to stay at home in case of non-severe symptoms. This information was widely relayed by the media. However, the patient had the possibility to go to the healthcare structure with a displacement certificate or could also call emergency medical services (SAMU) or SOS Médecins. During this period, a huge drop in the use of health care facilities by the population was observed, probably due to the fear of being infected by COVID-19 or in order to avoid saturation of health services. This decrease was also observed in emergency departments and SOS Médecins associations. Compared to 2019, the decrease was more marked for emergency departments visits than for SOSMed visits. 

In the manuscript, we added a sentence on the context during this special period, at the end of paragraph 1 of introduction section (see below, highlighted in yellow): 

“The first cases of the novel coronavirus disease 2019 (COVID-19) were reported in Wuhan, China, in December 2019 (1). On 30 January 2020, the World Health Organisation (WHO) declared COVID-19 as a public health emergency of international concern (2). In France, the first cases were confirmed on 24 January 2020. A national surveillance strategy was gradually implemented by Santé publique France (SpFrance), the French Public Health Agency, from 13 January 2020 (3). This surveillance formed part of the French national crisis management plan, organised in different phases. Phase 1 (from 2 January to 29 February 2020) involved the surveillance of individual cases and contact tracing in order to prevent the introduction of Sars-Cov-2 into the French territory. Phase 2 (from 1 to 13 March 2020) aimed to identify and break the chains of transmission in order to delay transmission in population. Phase 3 (since 14 March 2020) involved population surveillance to reduce the dissemination of the virus in the population and mitigate its impact in the health care system (4). From week 12 (16-22 March) to week 19 (4-10 May), a national lockdown was declared by the French government. Health authorities recommended to population to stay at home in case of non-serious symptoms. However, patient had the freedom to go to healthcare structure with a displacement certificate and could call emergency medical services (SAMU) or SOS Médecins.”

Comment 2#1. The fact that no unique identifier in the systems may be of concern, especially if duplicative counts were prevalent. What was the estimated rate of duplicates in counting the visits?

When data are received and integrated in our database, the visits with same information on the same date of consultation (same birth date, sex, city of residence and ED (or SOS Médecins associations)) are identified and considered as duplicates. Only one visit is integrated and the duplicates are deleted. The number of duplicates is very low (<1%). However, repeated visits on distinct days are counted separately.

The objective of syndromic surveillance is to measure the use of the healthcare, but not to track individuals or incidence rates. The surveillance is thus based on the count of visits instead of the monitoring of distinct persons. As the data are anonymous when sent to SpFrance, it is difficult to assess the proportion of these visits in the total number of visits.

We added more precisions in a paragraph of methods section of the manuscript, at the end chapter “Material” (see below, highlighted in yellow) : 

“Neither network attributes a unique identification number to patients. Because the goal of syndromic surveillance is to measure use of healthcare, surveillance is based on the count of visits instead of the monitoring of distinct persons. Repeated visits of a same patient on distinctive days are counted separately.” 

Comment 3#1. Did the left and right panels of Figure 1 use different normalization approaches? In the left panel it seems each age group has its own denominator, while in the right panel it seems the suspected cases were used as the single shared denominator across the ages groups so that the “All age group” percentage appears to be the sum of the others. Please clarify or consider using a more consistent way of normalizing.

In the left panel of figure 1, we showed the proportion of COVID-19 related visits among all coded visits by age group. For each age group, the numerator is the number of COVID-19 related visits and the denominator is the number of visits for all coded visits.

In the right panel of figure 1, we presented the proportion of the number of hospitalisations among COVID-19 related visits by age group. For each group, the numerator is the number of hospitalization after COVID-19 related visits and the denominator is the number of COVID-19 related visits. 

We have revised the title of Figure 1 as followed: 

“Figure 1: Proportion of daily COVID-19 related visits among overall coded visits (left) and proportion of daily hospitalisations among overall COVID-19-related visits (right), by age group in emergency departments and SOS Médecins associations in France (including overseas territories and Corsica) from 17 February to 28 June 2020.”

Comment 4#1. In Table 2 ED visits, please clarify why the column Distribution of COVID-19-related visits there is 0.0% Missing Gender, but in the column Proportion of hospitalisations following COVID-19-related visits there is 24.1% Missing Gender? How could that happen if the latter was a subset of the former?

The distribution of COVID-19 related visits (in percentage) for the category « missing gender” was obtained by dividing the number of COVID-19 related visits (29) by the number of overall COVID-19 visits (140,011). The result was 0.02% (with 1 decimal it is 0.0%). 

The proportion of hospitalisation after a COVID-19 related visit for this category was obtained by dividing the number of hospitalisation after a COVID-19 related visit (7) by the number of COVID-19 related visits (29). The result was 24.1%. 

We revised the title of the column of the proportion of hospitalization in Table 2 as follows: “Proportion of hospitalisations among COVID-19-related visits (%)”.

Comment 5#1. How did the baseline data transmission volumes change during the study period? It would be informative to visualize by using stacked area charts (especially with raw counts) that show the non-COVID-related visits at the bottom and the COVID-related visits stacked on top over time.

During the study period, data transmission and data quality remain stable and were not reduced by the emergent situation. Like in other countries (USA, Italy, UK)*, we observed a huge decrease of a majority of non-COVID-19 related visits, while COVID-19 related visits increased. 

*Lange SJ, Ritchey MD, Goodman AB and al., Potential indirect effects of the COVID-19 pandemic on use of emergency departments for acute life-threatening conditions - United States, January-May 2020. Am J Transplant. 2020 Sep;20(9):2612-2617. doi: 10.1111/ajt.16239. PMID: 32862556.

Stella F, Alexopoulos C, Scquizzato T, Zorzi A. Impact of the COVID-19 outbreak on emergency medical system missions and emergency department visits in the Venice area. Eur J Emerg Med. 2020 Aug;27(4):298-300. doi: 10.1097/MEJ.0000000000000724. PMID: 32618771.

Hughes HE, Hughes TC, Morbey R, Challen K, Oliver I, Smith GE, Elliot AJ. Emergency department use during COVID-19 as described by syndromic surveillance. Emerg Med J. 2020 Oct;37(10):600-604. doi: 10.1136/emermed-2020-209980. Epub 2020 Sep 18. PMID: 32948621; PMCID: PMC7503196.

 

We added a sentence on this observation in results (see below, highlighted in yellow). 

“Overall study period

The time course of COVID-19-related visits was concomitant across all age groups (Fig. 1). The highest proportions of COVID-19-related visits among overall visits were recorded among patients aged 45-64 years in the two networks as well as in the youngest adults (15-44 years) in SOSMed (Fig. 1). During the peak, one out of three visits was related to COVID-19 in these age groups. Among the elderly (65 years and older) during the peak, 23.7% (n=1,649) and 18.3% (n=220) of ED and SOSMed visits, respectively, were linked to COVID-19. In SOSMed, COVID-19-related visits of children (under 15 years) showed a similar pattern to those of adults, reaching 12.1% of overall visits at the peak, whereas in ED, these visits were limited during the entire study period and reached 3% of overall visits in late March (Fig. 1). 

Non-COVID-19 related visits in ED and SOSMed associations sharply decreased reaching their lowest level 2 weeks after the beginning of lockdown period. Beginning April 2020, a rebound of these visits was observed particularly in ED. “

The figure below show the temporal evolution of COVID-19 related visits and non-COVID-19 related visits in the two networks but due to different scales in raw count between the two indicators, COVID-19 related visits are hard to see. We propose not to add this figure to the manuscript. 

Figure: Number of COVID-19 and non-COVID-19 related visits in ED, from 17 February to 28 June 2020 in France Figure: Number of COVID-19 and non-COVID-19 related visits in SOS Medécins associations, from 17 February to 28 June 2020 in France

 

Comment 6#1. Was there a possible explanation for the higher rate of female COVID-19-related visits? For example, it would be informative to break down by age group distribution to see if there was additional caution in caring for pregnant women.

• In all coded ED visits, males usually consult more than females in all ages groups: under 15 years old (9.5% vs 7.8%), 15-44 years old (19.1% vs 17.0%), 45-64 years old (11.7% vs 9.8%) and 65-74 years old (5.1% vs 4.4%). When regarding COVID-19 ED related visits, females are more frequent than males in all ages (538% vs 46.1%) and for two age groups of 15-44 years old (21.9% vs 14.5%) and 45-64 years old (15.9% vs 14.1%). 

• In all coded SOSMed visits, females are more represented than males in all groups (except for the youngest (4.6% vs 4.9%)): 15-44 years old (24.7% vs 17.2%), 45-64 years old (10.5% vs 7.4%), 65-74 years old (3.9% vs 2.5%) and 75 years old and more (8.0% vs 4.0%). For COVID-19 SOSMed related visits, females are also more represented than males in all age groups (except the youngest (10.2% vs 11.2%): 15-44 years old (31.1% vs 22.4%) and 45-64 years old (13.4% vs 9.6%), 65-74 years old (3.5% vs 2.6%) and 75 years old and more (4.4% vs 3.1%). 

Our system doesn’t record the pregnancy status of women at each visit. But the distribution by age group detailed above showed that the higher rates of females COVID-19 related visits are not specific to young women. 

Our results are consistent with publications of COVID-19 surveillance in Veneto region (Italy) and Spain, which found higher rate of COVID-19 related visits but less severity in female. 

Different hypothesis could explain this higher rate (physiology, behavior about health care in the anxious context of this emergent pathology, …). Complementary studies could be necessary to explore this result. . 

In Discussion section, chapter “Results consistent with other data sources “, we added a sentence about hypothesis on sex differences in COVID-19 (see below, highlighted in yellow). 

“Results consistent with other data sources

Our observations are consistent with other data sources used for monitoring the outbreak in France; ED and SOSMed COVID-19 related visits were correlated to laboratory-confirmed tests, hospital and intensive care admissions. The population usually visiting SOSMed (younger than in ED) and location of associations (in urban settings) could partly explained the higher rate of children in SOSMed COVID-19 related visits. As reported in northeast France as well as in international studies, being male and advanced age were related to severe COVID-19 and death (14-17). COVID-19 surveillance in the US showed that 75.5% of recorded cases were aged 18-64 years and 52.1% were women, whereas among hospitalised patients with a confirmed laboratory test, 42.5% aged 65 years and older and 50.6% were men (18). Regarding the severity in male, as suggested in publications, it could be explained by different factors like immune system, sex hormones, physiological factors, sociocultural behaviours regarding health and underlying comorbidities (17, 19, 20).”  

Comment 7#1. A.What was the insurance mechanism for visiting SOSMed versus ED in France? B. Did the higher rate of children visiting SOSMed than ED indicate potential socioeconomic disparity? Please discuss.

A. In France, 70% of healthcare costs are covered by the national social insurance and the remained part are covered by private insurance. This coverage is the same for ED (which is a public structure) and SOSMed association (which is independent). However, an extra cost could be applied to SOSMed visits if consultation is done during out-of-hour. This extra-cost is fully charged to patient who will be reimbursed by his private insurance.

B. Usually, we observed higher rates of children visits in SOSMed than in ED, independently of COVID-19. Common respiratory infections (bronchitis, ORL infection, … ) or diseases (abdominal pain, gastro-enteritis, …) are the most frequent pathologies in SOSMed visits. This may highlight the fact that patients with mild symptoms prefer to seek care in ambulatory structure rather than going to the hospital where severe cases are more frequent and waiting time is longer. 

At the beginning of this emergent epidemic, the fear of contamination at hospital may have contributed to favor a SOSMed visit (which provides in-home visits), instead of going to an ED.

Another hypothesis could be due to the location of SOS Médecins associations (mainly located in urban areas). 

We have poor elements to analyze socioeconomic disparity through our data. This could be interesting to analyze in further studies. 

In discussion section, we added a sentence on this point (see below, highlighted in yellow) 

“Results consistent with other data sources

Our observations are consistent with other data sources used for monitoring the outbreak in France (laboratory-confirmed tests, hospital and intensive care admissions for COVID-19, COVID-19 related deaths). The population usually visiting SOSMed (younger than in ED) and location of associations (in urban settings) could partly explained the higher rate of children in SOSMed COVID-19 related visits.”

 

Minor comments:

Comment 1#1. The notion of “community” is highlighted in the title and several places. However, there lacks any contrast example of “non-community” surveillance to help readers appreciate the uniqueness of the SurSaUD system.

It seems that the term “community” is confusing and is not a correct translation. The system is not limited to persons who live in community (in contrast with non-community people); the surveillance system concerns the general population. 

We revised the title as following: “Syndromic surveillance: A key component of population health monitoring during the first wave of the COVID-19 outbreak in France, February-June 2020”.

The term has also been revised in the manuscript.

Comment 2#1. If the COVID-19 code set was evolving along with the proficiency that the coders gradually gained over time, how could the trends be interpreted reliably with such confounding effects?

We have described the evolution of the use of each ICD-10 code separately in this paper, in order to share this information with the scientific community, since large part of countries used also ICD-10 codes. But for the population surveillance in routine, COVID-19 related visits in ED were monitored by considering all ICD-10 codes together. ED physicians used the different ICD-10 codes progressively according to their availability. At the beginning, they used mainly the most generic codes. Then they could use either generic codes or more specific codes according to the health situation. 

We have also checked that the COVID-19 related visits for each ICD-10 code were not provided by a limited number of EDs and verified that EDs using each code were located in the whole territory (not only in one specific region). Such limitations, if existed, might introduce a confounding effect, according to the spread of the epidemic.

The observed epidemic trend is reliable and consistent with those obtained with the other data sources used to monitor the pandemic. We are confident with the fact that his trend reflected outbreak and not of the coding practices.

Comment 3#1. Were there any observed differences in the COVID-19 coding practice (e.g., distribution of common code usage) compared to other countries that also used ICD-10 (if published).

To our knowledge, few publications have described retrospectively COVID-19 related visits in ED during the first wave and these publications didn’t present the distribution of visits by ICD-10 codes. 

Comment 4#1. Please enhance the clarity when the word “department(s)” or “departmental” is mentioned by itself.

In France, “department” corresponds to an administrative geographical level (about 100 departments on the territory). In order to avoid confusion with ED, we have changed the terms in “districts” and have added a definition of this term when it is used the first time in the text. 

Comment 5#1. Please add a citation after mentioning of the reproduction number. 

We added reference of the national weekly bulletin published on 30 July 2020 and a publication on reproductive number methodology. 

Comment 6#1. The last part of Discussion involves a good amount of promotion materials, which could be reduced to allow for strengthening points around the study results.

We think that such syndromic surveillance system is the best solution for a reactive surveillance, particularly for emergent public health events in infectious field as well as for assessing the impact of environmental events or industrial accidents. This kind of events (particularly environmental events) could occur more frequently with climate change. The availability of such system when an emergent event occurs is fundamental to be reactive and to have a baseline with observed data the days/weeks/years before the event. Furthermore, such system is the only one that are able to monitor several simultaneous events (eg. COVID-19, mental disorders, other infectious diseases in winter period, asthma and usual pathologies in spring, heat-related consultations during heatwaves, …).

That is why it is essential for us to promote and encourage countries to implement such system.

We have revised the Discussion section and reduced it a little.

Comment 7#1. Since it has been a year, please briefly discuss whether (or not) the system still sustains and has been able to function consistently in surveillance of the later waves of the epidemic. 

SurSaUD® is still a part of the ongoing surveillance in complement to the other surveillance systems (information system for victim follow-up (SIVIC), surveillance in long-term care facilities (ESMS), testing information (SIDEP), contact tracing, and cluster monitoring). This provides a visibility of the impact on the health care system. 

In order to outline this point, we revised the last paragraph of Discussion section (see below, highlighted in yellow): 

“After a summer period marked by fewer COVID-19-related visits in the two networks, the second wave began again in September 2020 with a sharp increase in October. To control this second wave, national and local authorities implemented a series of mitigation measures at the local and national levels, including a second nationwide lockdown from 30 October to 15 December 2020 and curfews (25). ED and SOSMed in complement to the other sources (laboratory tests, long-term care facilities, critical care, … . ) were still used to monitor the epidemic during this second wave and further. Future studies will be conducted to compare the characteristics of COVID-19-related visits during the different waves.” 

 

Reviewer #2(#2): The authors report the epidemiological characteristics of those with a COVID-19-related visit captured by 2 health monitoring systems (emergency department visits and SOS Médecins) from February 17th to June 28th, 2020. Characteristics of this population during a national lockdown period between March 16th and May 10th were also described. COVID-19 related visits were categorized by any one of 9 specified COVID-19 specific ICD-10 codes. Results were also reported by geographic area in France. It was found that 4.0% of all ED visits and 5.6% of all SOSMed visits were COVID-19-related. The peak was observed in the nationwide lockdown period. The authors conclude that these syndromic surveillance systems allow for community-based health monitoring of COVID-19 in France.

This is a timely and relevant study that adds to the existing literature of syndromic surveillance tools in complementing laboratory based/PCR based surveillance of COVID-19. Figures and tables are clear. The authors adequately reported summary statistics and used appropriate methodologies with a few suggestions and some areas in need of improvement. Overall more can be added to highlight why this study is needed and how the results will be used.

Thank you for your appreciation.

Comment 1#2: Background: Details about each wave should be moved to a new first section in Methods which outlines key information about the study setting and context. Also for background section, please elaborate further on why this study needed to be done.

a. It was important for us to provide minimal details on the different phases of the French government's management of the crisis for a better understanding of the study context. Only period of lockdown was used for analysis. This is why we would like to keep stages description in the "context" section. 

b. For us, the study was important to publish since it contributes to:

• Share the epidemiological report of characteristics of patients who consult in these two data sources, which are available 7/7 days and cover the national population,

• Share the definition of the indicator used to monitor the direct impact of the epidemic, for further international comparison

• Share the assessment of the system in such exceptional events. This reactive early warning system remains reliable, stable, rapidly adaptable and can interact with physicians. That could help other countries to identify and implement such systems, that could be used for various emergent situations in infectious but also in environmental or industrial domains.

We add a sentence in Introduction section, paragrah 4 (please see below, highlighted in yellow). 

“This study describes the characteristics of COVID-19-related visits in EDs and SOSMed associations at the national and regional levels from 17 February to 28 June 2020, including the first nationwide lockdown. We highlighted first days of use of SyS to monitor such exceptional event especially in terms of reactivity, indicator’s design and implementation, interaction with physicians. That could help other countries to implement such surveillance for various emergent situations of health concern.”

Comment 2#2: Discussion- there is mention that the results are “consistent with other data sources used for monitoring the outbreak in France”- please tell us more about these other sources, and what the data presented here adds to what is already known. It would also be important to outline how the trends reported here compared to confirmed COVID-19 laboratory cases and confirmed hospitalizations. Finally, the discussion lacks a concluding paragraph which clearly highlights what this study has contributed.

a. The other data sources currently used for the population surveillance in France are the information system for victim follow-up (SIVIC), testing information system (SIDEP), surveillance in long-term care facilities (ESMS), contact tracing and cluster monitoring. 

During the surveillance, epidemic curve observed with Sys data was evaluated along with those observed in laboratory-confirmed tests (when the system was set up), hospital and intensive care admissions, COVID-19 related deaths. Other studies are on going on the whole data sources and are not published at this step. However, epidemiological bulletins were published every weeks during the epidemic, reporting the trends of each data source. 

We have added a sentence on ED and SOSMed COVID-19 related visits and data from other sources (lab test, hospital and intensive care admissions, deaths) and we have added the reference of our national SpFrance bulletin. 

“Results consistent with other data sources

Our observations are consistent with other data sources used for monitoring the outbreak in France (laboratory-confirmed tests, hospital and intensive care admissions for COVID-19, COVID-19 related deaths).”

b. Syndromic surveillance data are earlier than those of other system. It was the first to show the decline of epidemic curve, highlighting impact of governmental measures (lockdown, curfew, …). 

We add a paragraph on contribution of our study (see below, highlighted in yellow)

“Data analysis of SyS formed part (with other data sources) of the daily reports transmitted to decision-makers at the national and regional levels as well as the weekly national and regional bulletins published every Thursday on the SpFrance website (5). COVID-19-related visits in EDs were also used in complement with other data sources for different objectives: estimating the reproduction number R of the epidemic and providing criteria to determine the end of the lockdown. SyS data were also used by hospitals to monitor their occupancy rates and manage their needs of intensive care beds (24). 

This descriptive study provides a comprehensive picture of COVID-19 outbreak in emergency healthcare settings. It also highlighted how syndromic surveillance can be adapt to monitor rapidly a new emergent virus and to provide a real-time information to health authorities for decision-making.”

Specific suggestions:

Suggestion 1#2.) (Methods P4) “orientation after ED visit” should be changed to “destination after ED visit”

Thank you for your proposition. We have modified the expression as indicated. 

Suggestion 2#2.) (Methods P5) “All visits with a suspicion of COVID-19 recorded as PD or SD were selected for this study.” My understanding is visits that had one of the ICD-10 codes as either the PD or the SD were identified as a COVID-19-related visit. Please make this more clear and explicitly state that “COVID-19 related visits were categorized as those visits with at least 1 of the ICD-10 codes listed in table 1 as either the PD or the SD.”

Thank you for your proposition. The sentence has been modified as suggested. 

Suggestion 3#2.) (Methods P5) “Other diagnoses or symptoms such as cough, fever, respiratory failure, and dyspnoea were be coded as PD or SD in addition with one of the COVID-19-related codes.” Is there a reason why the presence of certain symptoms (without a qualifying ICD code) was not used in your definition of COVID-19-related visit? It is possible that a subset of patients may have only receive a diagnosis of “cough” and not one of the ICD-10 codes listed in table 1. Thus, some possible COVID-19-related visits may have been missed. If symptoms cannot be added to the definition of “COVID-19-related visit”, then a limitations paragraph should be included which states that this study was only able to identify potential COVID-19 visits that were assigned an ICD-10 code by a healthcare provider.

- See https://bmcpublichealth.biomedcentral.com/articles/10.1186/s12889-021-11303-9

which uses SyS in Canada and uses syndrome which were predefined groups of symptoms from ED to identify potential COVID-19 Cases

We need to monitor specifically COVID-19 related visits by considering the set of ICD10 codes and to monitor separately the other proxy-indicators based on diagnoses or symptoms such as cough, fever, respiratory failure, and dyspnea. In case of the occurrence of another epidemic with similar symptoms (like flu), it would be difficult to distinguish the respective trends of each epidemic. The monitoring of the other proxy indicators was particularly useful at the beginning of the epidemic, when the characteristics of this emergent pathology were not known and when ED/SOSMed didn’t have the national coding recommendations. Then the monitoring of these indicators was useful to identify reactively if another epidemic would occur. 

Yes, it’s possible that a subset of patients may have only receive a diagnosis of “cough” and not one of the ICD-10 codes listed in table 1, particularly in the couple of days at the beginning of the epidemic. However, we have provided coding recommendations both for COVID-19 related visits and for visits with proxy-symptoms and diseases, to all EDs and SOS Médecins associations rapidly, in order to reduce this risk.

Finally, the first objective of the surveillance system is not to count exhaustively the number of COVID-19 related visits, but to monitor the trends of the epidemics, in order to contribute to the management response.

A further analysis could be lead to estimate the excess numbers of visits for cough, fever or other symptoms, compared with a baseline determined with historical data. These excess numbers could be added to COVID-19 related visits with the hypothesis that these supplementary visits could be miscoded patients. 

At this step, we have revised the Discussion section, chapter on “Identification of COVID-19 related visits”, as you have suggested (see below, highlighted in yellow): 

“Identification of COVID-19-related visits 

In the two networks, patients suffering from COVID-19 were identified based on a clinical examination, since biological tests for SARS-COV2 infection were not available at the beginning of the study period. RT-PCR tests were progressively introduced from May 2020, although the results were poorly recorded in the system. This information was improved during the second wave of the epidemic following the development of rapid biological tests. However, doctors’ ability to identify patients with COVID-19 in EDs and SOSMed may have also improved with their better scientific knowledge of the disease. One example is the identification of dysgeusia as a symptom of this disease. 

We monitored specifically COVID-19 related visits by considering the set of ICD10 codes listed in Table 1. A subset of patients may have only diagnosed with proxy-indicators (cough, fever, respiratory failure and dyspnea) rather than one of the ICD-10 codes of interest; particularly in the couple of days at the beginning of the epidemic. However, we have provided coding recommendations both for COVID-19 related visits and for visits with proxy-symptoms and diseases, to all EDs and SOS Medecins associations rapidly, in order to reduce this risk.”

 Suggestion 4#2.) Be consistent with the lockdown period. In some areas of the manuscript, the March 16th - May 11th time period was used while in other the March 16th - May 10th time period was used.

Modification done. The period was 16 march to 10 may 2020. 

Suggestion 5#2.) Methods: Was the data pulled on one date at the end of the study period or was it extracted weekly/daily throughout the study period? Please specify this in the methods. Due to data back tracking, data can change depending on when it was extracted.

We extracted data at the end of study period. However, ED data are consolidated after 3 days and SOSMed data are consolidated in 1 or 2 days. 

For ED data, on average, 92% are transmitted within 24 hours, 99% within 48 hours and almost 100% in 72 hours. Missing data are updated up to 7 days. Based on consolidated data, 81% of these data have at least one medical diagnosis. The coding percentage is 75% at day +1, 79% at day+2 and 80% at day +3.

For SOS Médecins data, 97% are transmitted within 24 hours and 100% in 72 hours. Missing data are updated up to 3 days. The average coding percentage is 95% and is obtained at day+1. 

Consequently, there is no gap between analysis in routine and this retrospective analysis. 

We add in section method, precision on consolidation time (see below, highlighted in yellow) 

 “Data from emergency departments

Individual data are collected daily from computerised medical records completed during consultations in EDs in the OSCOUR® (Organisation de la surveillance coordonnée des urgences) network, which grew from 23 EDs in 2004 to around 700 in 2020. This system records 93.3% of all ED visits in France, varying from 85.6% to 100% depending on the region, including the French overseas territories (except Martinique). On average, 56,700 ED visits were recorded each day in 2019. Every morning, EDs transmit individual data from the previous 7 days to SpFrance. Most data (90%) are transmitted within 24 hours and consolidated within 72 hours.”

“Data from SOS Médecins

SOS Médecins is a network of emergency GP services providing emergency care in the private sector 24 hours a day, 7 days a week. They operate with hotlines that receive calls from patients, leading to medical advice, a home visit, or a consultation with a GP in a local SOSMed association. 

….

Every morning, SOSMed transmits individual data from the previous 3 days to SpFrance. Almost 97% of data are transmitted within 24 hours and consolidated within 72 hours.”

Suggestion 6#2.) (Results P17) “As in EDs, associated diagnoses were infrequent, with the most common being ENT diseases (rhinopharyngitis, angina) (24.1%). Please clarify how angina is considered an ENT disease?

According to ICD-10th version, angina is in the category “Other upper respiratory tract diseases (J30 to J39)” which are considered as “ENT infection”. 

Suggestion 7#2.) (Discussion P20) “On the contrary, we occasionally observed that visits for biological testing were miscoded (coded as U07.1 instead of U07.13, which was not included in our case definition).” This sentence may inadvertently been reversed- suggest rephrasing this as “ we occasionally observed that visits for biological testing were miscoded: instead of U07.1 they were coded as U07.13, which was not included in our definition.”

We don’t have taken into account your suggestion, since your proposition doesn’t correspond to the miscoded situation that we would like to discuss. We have reformulated the sentence, in order to clarify it:

“On the contrary, ED occasionally recorded COVID-19 related visits with the diagnosis U07.1, while the reason for the visits was the need to do biological testing because the patients (without clinical symptoms) were the contact of another person positive to COVID-19. The correct code for such visits would be U07.13 (which was not included in our case definition).” ________________________________________

---

## [Decision Letter · Decision Letter 1]

14 Sep 2021

PONE-D-21-12135R1Syndromic surveillance: A key component of population health monitoring during the first wave of the COVID-19 outbreak in France, February-June 2020PLOS ONE

Dear Dr. Thiam,

Thank you for submitting your manuscript to PLOS ONE. After careful consideration, we feel that it has merit but does not fully meet PLOS ONE’s publication criteria as it currently stands. Therefore, we invite you to submit a revised version of the manuscript that addresses the points raised during the review process.

Specifically, please improve the quality of your figures, and considered moving some figures to the Supplementary Materials as suggested by Reviewer 1.

We look forward to receiving your revised manuscript.

Kind regards,

Siew Ann Cheong, Ph.D.

Academic Editor

PLOS ONE

Journal Requirements:

Additional Editor Comments (if provided):

Reviewers' comments:

Reviewer's Responses to Questions

**Comments to the Author**

1. If the authors have adequately addressed your comments raised in a previous round of review and you feel that this manuscript is now acceptable for publication, you may indicate that here to bypass the “Comments to the Author” section, enter your conflict of interest statement in the “Confidential to Editor” section, and submit your "Accept" recommendation.

Reviewer #1: All comments have been addressed

Reviewer #2: All comments have been addressed

2. Is the manuscript technically sound, and do the data support the conclusions?

Reviewer #1: Yes

Reviewer #2: Yes

3. Has the statistical analysis been performed appropriately and rigorously? 

Reviewer #1: Yes

Reviewer #2: Yes

4. Have the authors made all data underlying the findings in their manuscript fully available?

Reviewer #1: Yes

Reviewer #2: Yes

5. Is the manuscript presented in an intelligible fashion and written in standard English?

Reviewer #1: Yes

Reviewer #2: Yes

6. Review Comments to the Author

Reviewer #1: The revision is responsive and has addressed the reviewer’s previous critiques.

Minor comments:

1. The figures read blurred in the PDF, making it difficult to see the details.

2. The stacked area charts of non-COVID and COVID-related visits (in the response letter) look informative. It could be included as supplemental results if the authors do not feel like putting them in the main manuscript.

3. The use of English could benefit from another round of proofread by a native speaker.

Reviewer #2: (No Response)

7. PLOS authors have the option to publish the peer review history of their article (what does this mean?). If published, this will include your full peer review and any attached files.

Reviewer #1: No

Reviewer #2: **Yes: **Arjuna S. Maharaj

---

## [Author Response · Author response to Decision Letter 1]

29 Oct 2021

Answers to reviewers

We thank the reviewers for the time spent on this new review and for the comments. We have taken into account all the recommendations. Detail of the answers to the comments are below (in bold). 

References review:

Comments: Please review your reference list to ensure that it is complete and correct. If you have cited papers that have been retracted, please include the rationale for doing so in the manuscript text, or remove these references and replace them with relevant current references. Any changes to the reference list should be mentioned in the rebuttal letter that accompanies your revised manuscript. If you need to cite a retracted article, indicate the article’s retracted status in the References list and also include a citation and full reference for the retraction notice.

As recommended by journal requirements, we used Vancouver style trough Endnote software for formatting our references. 

We also made some modifications in our list of references between the 1st and 2nd version of manuscript submitted. Please find below the changes made: 

- For reference 1, the website link was updated (1. WHO. Novel coronavirus China. Geneva: World Health Organization; 2020 [Available from: https://www.who.int/emergencies/disease-outbreak-news/item/2020-DON233.); 

- For reference 5, the website link was updated. The current link direct to SpFrance COVID-19 web page which is regularly updated (5. France Sp. Coronavirus (COVID-19) - Santé publique France 2021 [Available from: https://www.santepubliquefrance.fr/dossiers/coronavirus-covid-19)

- We have retracted the following reference : France Sp. Point épidémiologique hebdomadaire national [Available from: https://www.santepubliquefrance.fr/maladies-et-traumatismes/maladies-et-infections-respiratoires/infection-a-coronavirus/documents/bulletin-national/covid-19-point-epidemiologique-du-27-aout-2020. 

This reference was cited to highlight surveillance systems used to monitor COVID-19 outbreak in France. Instead of having a specific reference, pointing to a single bulletin, we preferred to cite the reference 5. France Sp. Coronavirus (COVID-19) - Santé publique France 2021 [Available from: https://www.santepubliquefrance.fr/dossiers/coronavirus-covid-19, which is SpFrance web page dedicated to COVID-19 with various information (data, statistics, bulletins) regularly updated. 

- Reference 13 was added to highlight utilization of other data used to monitor outbreak (laboratory-confirmed tests, hospital and intensive care admissions for COVID-19, COVID-19 related deaths) and the reproduction number R of the epidemic.

(13. France Sp. COVID-19 : point épidémiologique du 30 juillet 2020 France: Sante publique France; [updated 30 july 2020. Available from: https://www.santepubliquefrance.fr/maladies-et-traumatismes/maladies-et-infections-respiratoires/infection-a-coronavirus/documents/bulletin-national/covid-19-point-epidemiologique-du-30-juillet-2020.).

- References 14 to 16 were added to emphasize our discussion on sex comparison regarding COVID-19. 

14. Kaeuffer C, Le Hyaric C, Fabacher T, Mootien J, Dervieux B, Ruch Y, et al. Clinical characteristics and risk factors associated with severe COVID-19: prospective analysis of 1,045 hospitalised cases in North-Eastern France, March 2020. Eurosurveillance. 2020;25(48):2000895.

15. surveillance Wgft, Spain coC-i. The first wave of the COVID-19 pandemic in Spain: characterisation of cases and risk factors for severe outcomes, as at 27 April 2020. Eurosurveillance. 2020;25(50):2001431.

16. Stokes EK, Zambrano LD, Anderson KN, Marder EP, Raz KM, El Burai Felix S, et al. Coronavirus Disease 2019 Case Surveillance - United States, January 22-May 30, 2020. MMWR Morb Mortal Wkly Rep. 2020;69(24):759-65.

A. Comments to the Author

Reviewer #1: The revision is responsive and has addressed the reviewer’s previous critiques.

Thank you very much.

Minor comments:

1. The figures read blurred in the PDF, making it difficult to see the details.

Sorry for this blurred figures. The figures have now been exported in tiff format as recommended in the guidelines and should be in a better quality. We also used PACE tools to meet requirement journal’s for figure 3. 

2. The stacked area charts of non-COVID and COVID-related visits (in the response letter) look informative. It could be included as supplemental results if the authors do not feel like putting them in the main manuscript.

We added in supporting information the stacked area charts of non-COVID and COVID-related visits with the following captions: S1 Fig. Number of COVID-19 and non-COVID-19 related visits in ED and SOSMed associations, from 17 February to 28 June 2020 in France 

3. The use of English could benefit from another round of proofread by a native speaker.

An English speaker have proofread the current manuscript.

---

## [Editor Report · Decision Letter 2]

4 Nov 2021

Syndromic surveillance: A key component of population health monitoring during the first wave of the COVID-19 outbreak in France, February-June 2020

PONE-D-21-12135R2

Dear Dr. Thiam,

We’re pleased to inform you that your manuscript has been judged scientifically suitable for publication and will be formally accepted for publication once it meets all outstanding technical requirements.

Kind regards,

Siew Ann Cheong, Ph.D.

Academic Editor

PLOS ONE
---

## [Editor Report · Acceptance letter]

2 Feb 2022

PONE-D-21-12135R2 

Syndromic surveillance: A key component of population health monitoring during the first wave of the COVID-19 outbreak in France, February-June 2020 

Dear Dr. Thiam:

I'm pleased to inform you that your manuscript has been deemed suitable for publication in PLOS ONE. Congratulations! Your manuscript is now with our production department. 

Kind regards, 

on behalf of

Dr. Siew Ann Cheong 

Academic Editor

PLOS ONE